# Stable isotope analysis of multiple tissues from Hawaiian honeycreepers indicates elevational movement

**Kristina L. Paxton** [1] *, **Jeffery F. Kelly** [2,3], **Sandra M. Pletchet** [2], **Eben H. Paxton** [4]

**1** Department of Biology, University of Hawai'i Hilo, Hilo, HI, United States of America, **2** Corix Plains Institute, University of Oklahoma, Norman, OK, United States of America, **3** Department of Biology, University of Oklahoma, Norman, OK, United States of America, **4** U.S. Geological Survey Pacific Island Ecosystems Research Center, Hawai'i National Park, HI, United States of America

* kpaxton@hawaii.edu

## Abstract

We have limited knowledge of the patterns, causes, and prevalence of elevational migration despite observations of seasonal movements of animals along elevational gradients in montane systems worldwide. While a third of extant Hawaiian landbird species are estimated to be elevational migrants this assumption is based primarily on early naturalist's observations with limited empirical evidence. In this study, we compared stable hydrogen isotopes ($\delta^2$H) of metabolically inert (feathers) and active (blood plasma, red blood cells) tissues collected from the same individual to determine if present day populations of Hawaiian honeycreepers undergo elevational movements to track areas of seasonally high flower bloom that constitute significant food resources. We also measured stable carbon isotopes ($\delta^{13}$C) and stable nitrogen isotopes ($\delta^{15}$N) to examine potential changes in diet between time periods. We found that the majority of 'apapane (*Himatione sanguinea*) and Hawai'i 'amakihi (*Chlorodrepanis virens*) captured at high elevation, high bloom flowering sites in the fall were not year-round residents at the capture locations, but had molted their feathers at lower elevations presumably in the summer after breeding. $\delta^2$H values of feathers for all individuals sampled were higher than blood plasma isotope values after accounting for differences in tissue-specific discrimination. We did not find a difference in the propensity of elevational movement between 'apapane and Hawai'i 'amakihi, even though the 'amakihi is considered more sedentary. However, consistent with a more generalist diet, $\delta^{15}$N values indicated that Hawai'i 'amakihi had a more diverse diet across trophic levels than 'apapane, and a greater reliance on nectar in the fall. We demonstrate that collecting multiple tissue samples, which grow at different rates or time periods, from a single individual can provide insights into elevational movements of Hawaiian honeycreepers over an extended time period.

**Data Availability Statement:** All data is available at a USGS data repository called ScienceBase-Catalog with the following citation and url: Paxton KL, Kelly JF, Pletchet SM, Paxton EH. 2020. Hawaii

Volcanoes National Park stable isotope values from Hawaii forest birds 2012. U.S. Geological Survey data release: https://doi.org/10.5066/P98I4EP7.

**Funding:** Funding for this study was provided to EHP through a U.S. Geological Survey Natural Resource Preservation Project grant. KLP was supported by a National Science Foundation (NSF) Centers for Research Excellence in Science and Technology (CREST) grant (0833211). JFK was supported by NSF grant EF-1840230. Any opinions, findings, and conclusions or recommendations expressed in this material are those of the authors and do not necessarily reflect the views of NSF. Any use of trade, firm, or product names is for descriptive purposes only and does not imply endorsement by the U.S. Government. The funders had no role in the study design, data collection and analysis, decision to publish, or preparation of the manuscript.

**Competing interests:** The authors have declared that no competing interests exist.

## Introduction

The movement of animals in response to seasonal fluctuations in climate and availability of resources is a widespread and taxonomically diverse behavior occurring in animals such as birds, bats, insects, and ungulates [1, 2]. However, our understanding of ecological factors shaping migratory behavior has primarily been shaped by studies of long-distance migration across latitudinal scales. In contrast, we have limited knowledge of the patterns, causes, and prevalence of elevational migration despite observations of seasonal movements of animals along elevational gradients in montane systems worldwide [3, 4]. Unlike obligate, long-distance migration (i.e., every year the entire population migrates), elevational migration typically occurs over short distances and is facultative such that individuals may adapt their behavior in response to environmental conditions that vary from year to year [3]. Seasonal migration across elevations may be driven by ecological factors such as spatial and temporal variation in food resources, weather events, or predation risks that vary at different elevations [4]. For example, the abundance of some frugivorous and nectarivorous bird species, birds that primarily eat fruit and nectar, respectively, in tropical montane systems has been shown to vary across elevations associated with seasonal changes in fruit and flower availability [5, 6]. Alternatively, elevational migration can be in response to storms, such as seen with white-ruffed manakins (*Corapipo altera*) in montane wet forests in Central America that migrate to lower elevations following severe storm events as a result of reduced foraging opportunities at high elevations [7].

While elevational migration has been well documented for Neotropical frugivore and nectarivore bird species, many nectarivores outside of the Neotropics, along with other feeding guilds (e.g., insectivores, granivores), are also thought to engage in elevational migrations [3]. For example, a third of extant Hawaiian landbird species are estimated to be elevational migrants [3]. However, for many Hawaiian species this assumption is primarily based on early naturalist's observations of Hawaiian forest birds making long, high flights over the forest canopy, seasonal changes in the abundance of birds at different elevations [8–11], and observations of birds in low-elevation habitat following large storms [8]. Early naturalists hypothesized that nectarivorous birds made seasonal movements across elevations as they tracked the timing of flowering 'ōhi'a (*Metrosideros polymorpha*). 'Ōhi'a is the dominant tree from sea level to tree line in Hawaiian wet tropical forests, accounting for more than 80% of the biomass of native forests [12], and is the primary nectar resource utilized by Hawai'i's native nectarivorous birds [13]. Differences in the timing of 'ōhi'a bloom by 'ōhi'a varieties that occur at different elevations creates spatially and temporaly variable distributions of bloom [14, 15] that may drive elevational migration. However, this "Elevational-Migration Hypothesis" is based on limited empirical evidence of seasonal movements of Hawaiian forest birds across elevations (but see [16]) given the challenges in following birds across Hawai'i's rugged and remote tropical forests. Moreover, recent correlative studies examining the seasonal abundance of Hawaiian forest birds at different elevations have not found strong synchrony between bird abundances and the flowering phenology of 'ōhi'a [17, 18]. The lack of correspondence between flower and bird densities indicates that elevational movements may not be as prevalent as early naturalists thought or that present day movement strategies of native Hawaiian forest birds are potentially changing because of factors such as the loss and fragmentation of forests across the landscape [19], reduced competition for nectar resources as populations decline or become extinct [20], or higher disease prevalence of introduced diseases (e.g., avian malaria, pox, mange) at low elevations [21].

Limitations in following individuals through time in steep mountainous terrains, particularly small species that are too light to carry tracking devices [22], has hindered our ability to

understand elevational movements in birds worldwide. The use of stable isotopes has been one approach to documenting movement of small animals across large geographic areas given that stable isotopes vary predictably across the landscape and are incorporated into animal tissues through biochemical processes [23]. Stable hydrogen isotopes ($\delta^2$H) in particular have become a well-established technique for studying long-distance movements of birds at continental scales (reviewed in [23]), but only a few studies in comparison have used $\delta^2$H to study elevational migration [24–28]. However, $\delta^2$H values vary predictably with not only latitude but also elevation, with approximately a 1‰ to 4‰ decrease of $\delta^2$H in precipitation with every 100 m increase in elevation [29]. Depletion of H with altitude results from Rayleigh distillation and depletion of precipitation as an air mass rises over a mountain range and loses moisture to rainfall and decreasing temperatures [29, 30]. Patterns of $\delta^2$H in precipitation are correlated with animal tissues as a result of biochemical processes [23], and thus animal tissues are expected to reflect the isotopic signatures of the elevation of feeding where the tissue was grown.

Stable carbon isotopes ($\delta^{13}$C) can also be used to understand elevational movement, although the gradient of change across elevations is much smaller, and thus, $\delta^{13}$C are often not as informative as $\delta^2$H for small elevational gradients. For example, the rate of increase in $\delta^{13}$C values of bird feathers was only ~1.3 to 1.5‰ per 1,000 m for adult male black-throated blue warblers (*Setophaga caerulescens*) and multiple hummingbird species collected across elevational gradients in the Appalachian and Andean mountains, respectively [24, 31]. Therefore, in areas with small elevational gradients, $\delta^{13}$C, along with stable nitrogen isotopes ($\delta^{15}$N), are better for providing an understanding of the diet niche of a species than movement [23].

Animal tissues incorporate local isotopic signatures over different time periods, from days to years [32–34], and thus at a single point in time different tissues can provide different time frames associated with the movement of animals. For example, feathers are metabolically inert after formation, and reflect the diet and water inputs of the bird only during the discrete period of feather growth (but see [35]). In contrast, metabolically active tissues like blood, liver, and muscle continuously incorporate the isotopic signature of their environment at varying rates. Blood plasma quickly incorporates isotopic signatures of the local environment with an average residency time of only ~3 to 5 days, while red blood cells (RBCs) and muscle have slower isotopic incorporation rates and integrate local isotopic signatures over longer timescales ranging from 1 to 2 months [32, 33, 36, 37]. However, inter-tissue differences in stable isotope values can not only be a function of differences in the residency time of isotopes, but also physiological mechanisms that control isotopic discrimination among different tissues resulting in tissue-specific discrimination [34]. Field and laboratory studies of birds have shown that feathers are more enriched in $^2$H than other tissues (e.g., plasma, RBCs, muscle), while isotopic discrimination of $^2$H between RBCs and blood plasma was not shown to differ [32, 33, 38, 39]. While controlled laboratory studies are beginning to shed light on the patterns of tissue-specific discrimination of stable isotopes, the processes that determine these patterns (e.g., differences in protein synthesis and nutrient routing between tissues) are still not completely understood [32]. Thus, comparisons of multiple tissues collected from the same individual at one point of time can provide insights into shifts in elevational movement and diet for different time periods of the annual cycle, as long as comparisons are made while also incorporating tissue-specific discrimination factors.

We tested the elevational-migration hypothesis in nectarivorous birds on the east side of Hawaiʻi Island in a wet tropical forest to determine if present day populations undergo elevational movements to access ʻōhiʻa flower blooms that form significant food resources. We captured birds at high elevation sites near the upper limits of the forest that had heavy ʻōhiʻa bloom. Birds were captured in the fall after peak molt (i.e., summer) and peak breeding (i.e.,

January to May) seasons were complete. To assess the propensity of movement, we compared $\delta^2$H of metabolically inert (feathers) and active (blood plasma, RBC) tissues collected from the same individual. Based on the elevational-migration hypothesis, we predicted that birds making elevational migrations upslope to high bloom areas would have lower blood plasma $\delta^2$H values, representative of the high elevation capture location, compared to their feathers that were grown at lower elevation breeding and molting sites. In contrast, if birds captured at high elevation, high bloom areas were resident breeders, we would expect similar stable isotope values among all tissue types, after accounting for differences in tissue-specific discrimination. We tested these predictions for two Hawaiian honeycreeper species, ʻapapane (*Himatione sanguinea*) and Hawaiʻi ʻamakihi (*Chlorodrepanis virens*), that vary in their degree of nectarivory and hypothesized propensity for movement. ʻApapane are nectarivores, primarily feeding from ʻōhiʻa flowers, but are also known to consume foliage arthropods during the breeding season [13, 40]. In contrast, Hawaiʻi ʻamakihi are generalists that eat foliage arthropods but also consume large quantities of nectar when available [41]. Hawaiʻi ʻamakihi are thought to be more sedentary than ʻapapane given differences in their foraging strategies and greater genetic structure [42], and thus we predicted that a greater proportion of ʻapapane captured at high elevation, high bloom areas would be elevational migrants making long-distance movements in search of ʻōhiʻa nectar. To examine potential changes in diet between time periods of the annual cycle we also measured $\delta^{13}$C and $\delta^{15}$N from feathers and RBCs of target birds. We predicted that Hawaiʻi ʻamakihi would have greater inter-tissue variation for both $\delta^{13}$C and $\delta^{15}$N given their more generalist diet compared to ʻapapane. However, we did not predict large differences in $\delta^{13}$C associated with elevational movements given the small elevational range (~500 to 1500 m) of movements expected by ʻapapane and Hawaiʻi ʻamakihi.

## Methods

### Ethics statement

The research conducted for this study was carried out in accordance with the Ornithological Council's guidelines for the use of wild birds in research and was approved by The University of Hawaiʻi at Hilo's Institutional Animal Care and Use Committee (protocol # UH 12–1315). Other permits were from the United States Department of the Interior bird banding laboratory (permit # 23064), Hawaii State Protected Wildlife Research Permit (WL 13–07), and National Park Service Research Permit (HAVO-2012-SCI-0041).

### Study species and area

We sampled ʻapapane and Hawaiʻi ʻamakihi at Hawaiʻi Volcanoes National Park in the upper Kaʻū Forest at three sites ranging in elevation from 1615 to 2191 m (Fig 1). Both species are Hawaiian honeycreepers (Fringillidea) that are locally abundant and widely distributed within the Kaʻū Forest and have been detected through surveys during the breeding season at elevations ranging from tree line (~2200m) to 700 m, with moderately high densities below 1500 m where mosquitoes that vector avian malaria are present [43]. The Kaʻū Forest is one of the largest intact native tropical wet forests on Hawaiʻi Island located on the southeast windward slopes of Mauna Loa Volcano. The forest is comprised primarily of mature ʻōhiʻa and varying amounts of koa (*Acacia koa*) with a predominantly native understory. Southern regions of the Kaʻū Forest have been used for cattle ranching over the last 150 years and consist of forested pastures with isolated patches of ʻōhiʻa-koa forests and understory grasses [43]. All three capture sites in this study were near tree line in low stature scrubby ʻōhiʻa patches that had heavy ʻōhiʻa bloom, immediately above the high stature mixed ʻōhiʻa-koa forest. We also attempted to capture birds in the high stature mixed ʻōhiʻa-koa forest which had little to no ʻōhiʻa bloom,

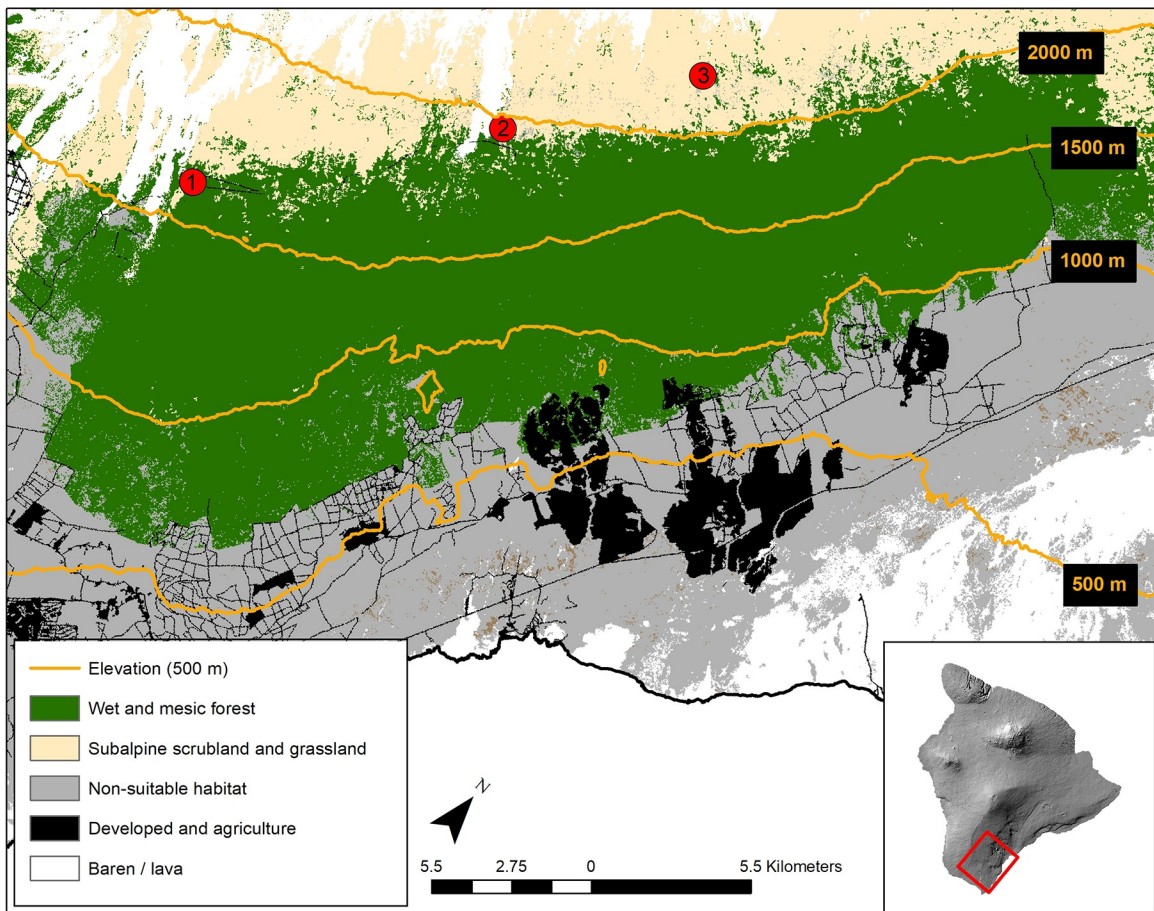

**Fig 1. Map of the Kaʻū Forest on Hawaiʻi Island displaying the capture sites at high elevations (depicted by contour lines).** Habitat layers were obtained from the Landfire program (https://www.landfire.gov).

however, we did not catch any birds most likely because densities of birds were so low. The climate of the Kaʻū Forest is affected by Mauna Loa Volcano as winds are driven around and upward creating three rainfall patterns: trade wind and thermally driven sea breeze cycles dominate rainfall patterns from Pāhala to Nāʻālehu, a rain-shadow is present in the area southwest of Kīlauea summit, and high elevation areas that are above the trade wind inversion zone have rainfall only during storms [44].

## Sample collection

We captured birds via passive mist netting at three high elevation sites that were experiencing high ʻōhiʻa bloom during the months of September and October in 2012. Upon capture, birds were banded with a U.S. Geological Survey (USGS) aluminum numerical band, weighed to nearest 0.1g with an electronic scale, aged and sexed based on plumage, breeding characteristics, and size (as described in [45]), and assessed for active molt. Additionally, we pulled the two outer most tail feathers and collected a blood sample via brachial vein for $\delta^2$H (plasma and RBCs) and $\delta^{13}$C and $\delta^{15}$N (RBCs only) analysis. For most birds, the amount of blood collected was enough for $\delta^2$H and $\delta^{13}$C and $\delta^{15}$N analysis. However, we prioritize blood for $\delta^2$H analysis when the blood volume was too low for both analyses to be conducted. We stored blood

samples on ice until the plasma and RBCs could be seperated at the end of the day, and then placed on dry ice until they could be transfered to a -20˚C freezer.

We used plasma samples to represent the isotopic signature of the capture location given the short residency time of blood plasma, while feather samples represented the location of molt. Molt typically occurs following the breeding season during primarily the summer and early fall, and is believed to occur largely on the breeding grounds [46, 47]. While Hawaiian honeycreepers can breed anytime between October to May, depending on weather conditions, their peak breeding season occurs between January to May [46]. RBCs represented the time period in between breeding and capture in the fall.

## Stable isotope analysis

Feather and blood samples were prepared for stable isotope analysis at the University of Oklahoma using the protocols outlined in [48]. Briefly, feather samples were cleaned with a 2:1 chloroform methanol solution as well as a phosphate-free detergent and rinsed in deionized water before drying for 24–36 hours under a fume hood. Plasma and RBCs were freeze-dried and powdered. Lipids were not extracted from blood samples prior to freeze drying given the low concentration of lipids in bird blood [49]. Feather material from the distal end of the sample, powdered RBC samples, and powdered plasma samples were weighed ($\delta^2H$: 140 to 160 µg, $\delta^{13}C$ and $\delta^{15}N$: 350 µg) and wrapped in a silver ($\delta^2H$) or tin ($\delta^{13}C$ and $\delta^{15}N$) capsule.

Samples were analyzed for stable hydrogen isotope ratios at the Colorado Plateau Stable Isotope laboratory (CPSIL; Flagstaff, Arizona, USA) using a comparative equilibrium approach with calibrated keratin standards to correct for uncontrolled isotope exchange between non-carbon-bound hydrogen in feathers and ambient water vapor [50]. The three calibrated isotope keratin standards analyzed with feather samples included: Cow Hoof (CBS; $\delta^2H$ = -197‰); Kudi Horn (KHS; $\delta^2H$ = -54.1‰); Spectrum Keratin Powder Lot SJ (SKP; $\delta^2H$ = -121.6‰). Stable hydrogen isotope ratios were determined with a Thermo Scientific Delta Plus isotope ratio mass spectrometer connected to a Thermo Scientific TC/EA elemental analyzer and configure through a Thermo Scientific CONFLO IV for automated continuous-flow analysis. Samples were analyzed for stable carbon and nitrogen isotope ratios at the University of Arkansas using a Thermo-Finnigan DeltaPlus isotope ratio mass spectrometer connected to an Carlo Erba elemental analyzer. Two standards analyzed with feather and RBC samples included USGS40 and BHCO (powdered brown-headed cowbird (*Molothrus ater*) feather) used extensively as a lab standard as documented by Kelly et al. [51]. Stable isotope ratios are expressed in standard notation, where $\delta^2H$, $\delta^{13}C$, and $\delta^{15}N$ = [(isotope ratio$_{sample}$/isotope ratio$_{standard}$)− 1] x 1000. Consequently, $\delta^2H$, $\delta^{13}C$, and $\delta^{15}N$ are expressed in parts per thousand (‰) deviation from a standard ($\delta^2H$: Vienna Standard Mean Ocean Water, $\delta^{13}C$: Vienna Pee Dee Belemnite, $\delta^{15}N$:Air). Measurement of the three keratin reference materials corrected for linear instrumental drift were both accurate and precise with mean $\delta^2H$ ± standard deviation of -198.0 ± 0.4‰ (CBS), -55.8 ± 0.7‰ (KHS), and -120 ± 1.8‰ (SKP). Likewise, repeated analysis of $\delta^{13}C$ and $\delta^{15}N$ standards were -26.2 ± 0.3‰ ($\delta^{13}C$ USGS40), -4.2 ± 0.3‰ ($\delta^{15}N$ USGS40), -15.7 ± 0.1‰ ($\delta^{13}C$ BHCO), and 7.6 ± 0.1‰ ($\delta^{15}N$ BHCO). We ran standards and a replicate sample every 10th sample and flagged any replicate sample that differed by > 6‰.

## Statistical analysis

We conducted all statistical analyses in R version 3.6.0 (R Core Team 2019) using the packages lme4 [52], lmerTest [53], and lsmeans [54]. We used general linear mixed models (GLMM) to separately examine differences in $\delta^2H$, $\delta^{13}C$, and $\delta^{15}N$ values among tissue types ($\delta^2H$: feathers, RBC, plasma; $\delta^{13}C$, $\delta^{15}N$: feathers, RBC) and species ('apapane, Hawai'i 'amakihi). Individual

was included as a random effect in each model to account for multiple tissues collected from the same individual. We assumed statistical significance at alpha ≤ 0.05, but with multiple comparisons we conducted a Tukey's post-hoc analysis of least squared means to determine differences among significant factors. Prior to running GLMMs we adjusted $\delta^2H$ feather values by -19.8‰ to account for tissue-specific discrimination between feathers and blood [34]. We adjusted $\delta^2H$ feather values based on the average difference in tissue discrimination factors of feathers and blood plasma (-18.6‰±4.4‰) and feathers and RBCs (-21‰±1.5‰) calculated from an experiment with house sparrows (*Passer domesticus*), the species most closely related to Hawaiian honeycreepers with tissue-specific discrimination values [38]. RBCs and blood plasma do not differ in isotopic discrimination of $^2H$; therefore, we did not adjust these tissue types [32, 33, 38, 39]. We did not adjust $\delta^{13}C$ and $\delta^{15}N$ feather values prior to analysis because tissue-specific discrimination factors for $\delta^{13}C$ and $\delta^{15}N$ are highly sensitive to diet [36, 55] and discrimination factors have not been established for a nectarivore. In addition, published discrimination factors have primarily been calculated for feathers and whole blood, but not RBCs [36, 55, 56]. Instead, similar to the approach of Podlesak et al. [36] we examined only general changes in diet by considering differences in $\delta^{13}C$ and $\delta^{15}N$ values between feathers and RBCs greater than 2‰ to be indicative of a change in diet.

We estimated the elevation of feather growth for feather samples collected at our study sites based on the relation between elevation and stable hydrogen isotopes in precipitation ($\delta^2H_p$) for wet tropical forests on the east side of Hawai'i Island, $\delta^2H_p$ = -0.018(elevation) -11.25. We derived the relationship between $\delta^2H_p$ and elevation using published volume-weighted average $\delta^2H_p$ values collected from 50 locations sampled across east Hawai'i Island, including sample locations within the Ka'ū Forest, and ranging in elevation from 6 to 4000 meters ([57] Appendix 1). Scholl et al. [57] calculated volume-weighted average $\delta^2H_p$ values for each location based on rainfall samples collected at 6-month intervals between August 1991 to August 1994. Because all three rain patterns (i.e., trade winds, rain shadow, high elevation) are present in the Ka'ū Forest we used all sampling locations representing these rainfall patterns to establish the relationship between $\delta^2H_p$ and elevation for the Ka'ū Forest. Prior to estimating elevation, we adjusted stable hydrogen isotopes in feathers ($\delta^2H_f$) to reflect $\delta^2H_p$ utilizing a conversion equation from a species in the same foraging guild as our study species, Rufous hummingbird (*Selasphorus rufus*). As nectarivores, hummingbirds and Hawaiian honeycreepers receive most of their hydrogen from plant-derived water and plant-derived carbohydrates through consumption of large quantities of nectar. Thus, in the absence of a species-specific conversion equation for our species [58], hummingbirds likely provide the best approximation of the relation between feather and precipitation $\delta^2H$ values for Hawaiian honeycreepers. The equation, $\delta^2H_p$ = 1.15($\delta^2H_f$) + 29.01, was calculated based on Rufous hummingbird feathers grown at known locations across North America [59]. The same relation between $\delta^2H_f$ and $\delta^2H_p$ was also found for Ruby-throated hummingbirds (*Archilochus colubris*) [60], indicating a consistent relation between $\delta^2H_f$ and $\delta^2H_p$ for nectarivorous birds.

## Results

We captured a total of 102 birds during the 2-month time period ($\delta^2H$ analysis: 'apapane = 60, 'amakihi = 42; $\delta^{13}C$ and $\delta^{15}N$ analysis: 'apapane = 51, 'amakihi = 36). We found significant differences in $\delta^2H$ values among tissues taking into account individual variability ($F_{2,118.3}$ = 172.5, p<0.001) with patterns not differing between species ($F_{1,99.7}$ = 1.6, p = 0.21). Feathers had the highest mean $\delta^2H$ values followed by RBCs, and then blood plasma (Fig 2). The average difference in $\delta^2H$ values between feathers and blood plasma collected from the same individual was 40.7‰ (range 24.0–62.5‰), suggesting feathers were grown at lower elevations

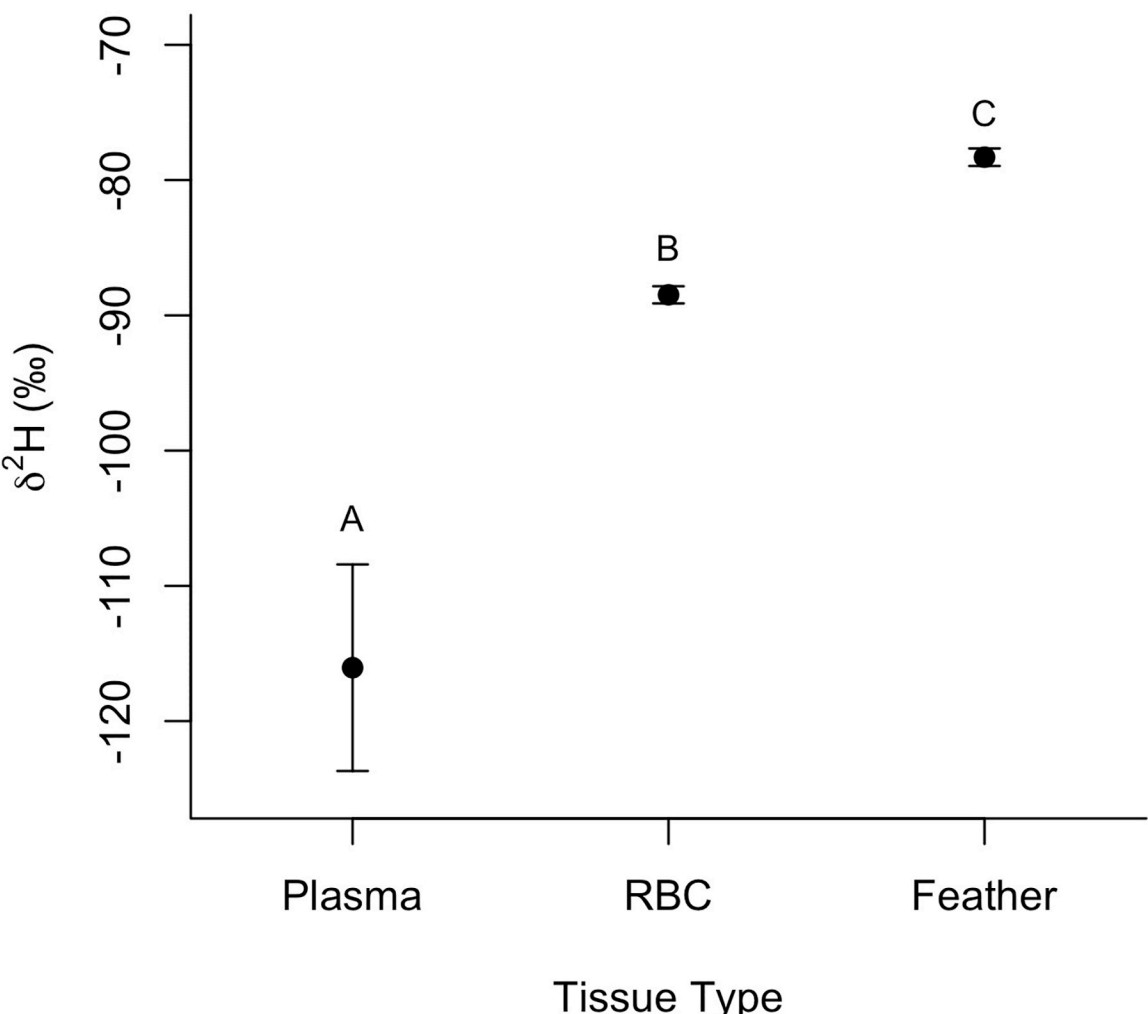

**Fig 2. Stable hydrogen isotope ($\delta^2$H) values (± SE) for plasma, red blood cells (RBC), and feathers adjusted for tissue discrimination collected from ʻapapane and Hawaiʻi ʻamakihi in the upper Kaʻū Forest.** Letters above bars indicate tissue types that are significantly different from one another based on a Tukey's post-hoc analysis.

than the capture location represented by the blood plasma sample. The average difference in $\delta^2$H values between RBCs and blood plasma of the same individual was 32.3‰ (range 23.9–41.4‰).

Based on the relationship between $\delta^2H_p$ and elevation on the east side of Hawaiʻi Island we found the average estimated elevation of feather growth (1501 m, range 505 to 2528 m) was lower than the elevation of all three capture locations (1615 m, 1968 m, 2191 m) (Fig 3). The average distance between the elevation of a birdʻs capture location and estimated elevation of feather growth was 562 m (range 5–1686 m) with the distance between capture and molt elevation increasing with increasing elevation of the capture location (Table 1). All of the honeycreepers captured at our highest elevation site were estimated to have grown their feathers at lower elevations with on average 1008 m (range 193–1686 m) between the capture site and the estimated molt location. In contrast, 83% and 45% of honeycreepers captured at the other two sites (1968 m, 1615 m) had an estimated molt location lower than their capture location.

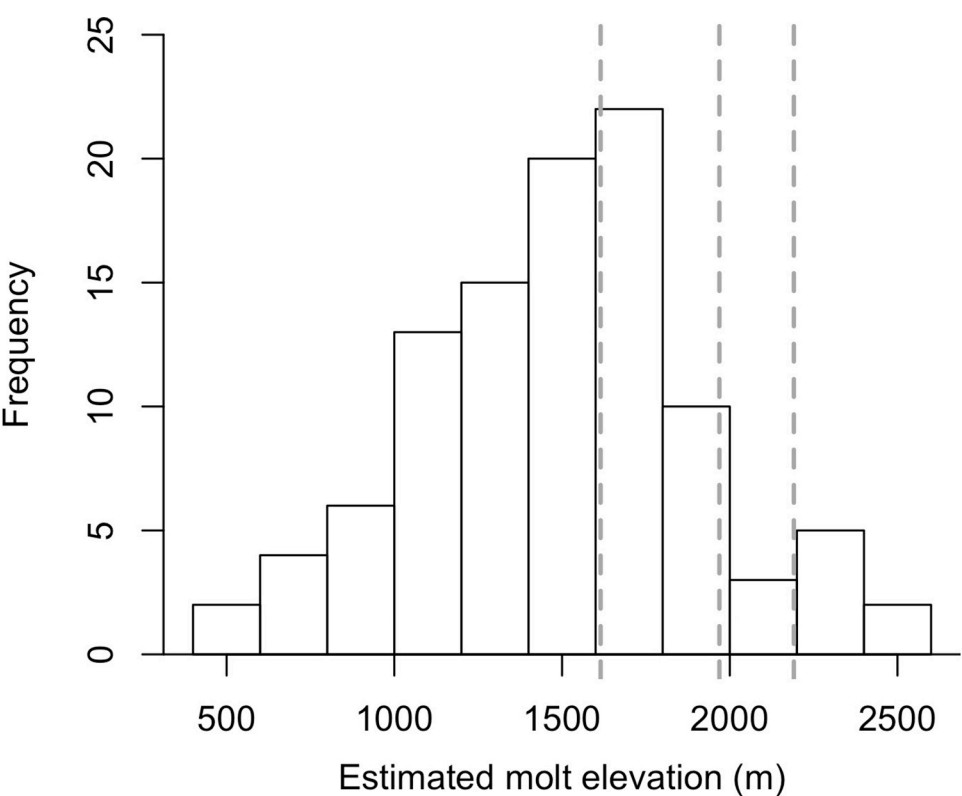

**Fig 3. Estimated elevation of feather growth for Hawaiʻi ʻamakihi and ʻapapane captured during the fall in upper Kaʻū Forest.** Dashed lines represent the elevations of the three capture sites.

We found significantly higher $\delta^{13}$C and $\delta^{15}$N values of feathers compared to RBC values ($\delta^{13}$C: $F_{1,86} = 161.4$, p<0.001, $\delta^{15}$N: $F_{1,86} = 30.3$, p<0.001) with overall higher $\delta^{13}$C and $\delta^{15}$N values for Hawaiʻi ʻamakihi compared to ʻapapane ($\delta^{13}$C: $F_{1,85} = 36.5$, p<0.001, $\delta^{15}$N: $F_{1,85} = 4.2$, p = 0.04) (Fig 4). However, for ʻapapane the average difference in isotope values between feathers and RBCs collected from the same individual was less than 2‰ ($\delta^{13}$C: 0.9‰, range -0.3 to 2.6‰, $\delta^{15}$N: 1.3‰, range -3.0 to 4.0‰) (Fig 5). In contrast, for Hawaiʻi ʻamakihi almost half ($\delta^{13}$C: 42%, $\delta^{15}$N: 44%) of the individuals captured had differences in isotope values between feathers and RBCs greater than 2‰ ($\delta^{13}$C: 2.0‰, range -1.9 to 3.6‰, $\delta^{15}$N: 2.6‰, range -3.2 to 8.2‰) (Fig 5).

## Discussion

We demonstrate that collecting multiple tissue samples, which grow at different rates or time periods, from a single individual can provide insights into elevational movements over an

**Table 1. For each capture location, the sample size (n) and mean, minimum, and maximum difference in elevation between a bird's capture location and their estimated molt location.** Site numbers refer to the locations indicated in Fig 1.

| Site | Elevation (m) | n | Difference in elevation of capture and estimated molt location | | |
|---|---|---|---|---|---|
| | | | Mean (m) | Minimum (m) | Maximum (m) |
| 1 | 1615 | 33 | 231 | 9 | 913 |
| 2 | 1968 | 35 | 442 | 5 | 1248 |
| 3 | 2191 | 34 | 1008 | 193 | 1686 |

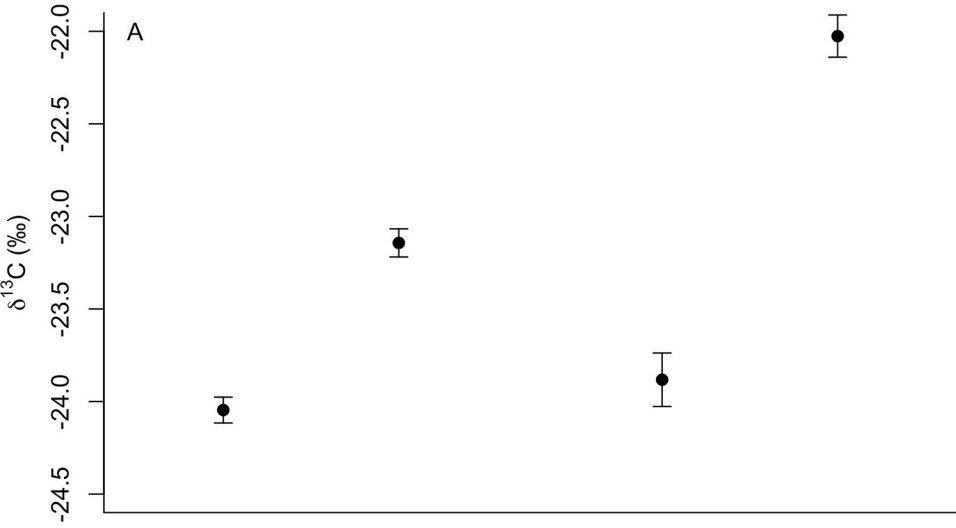

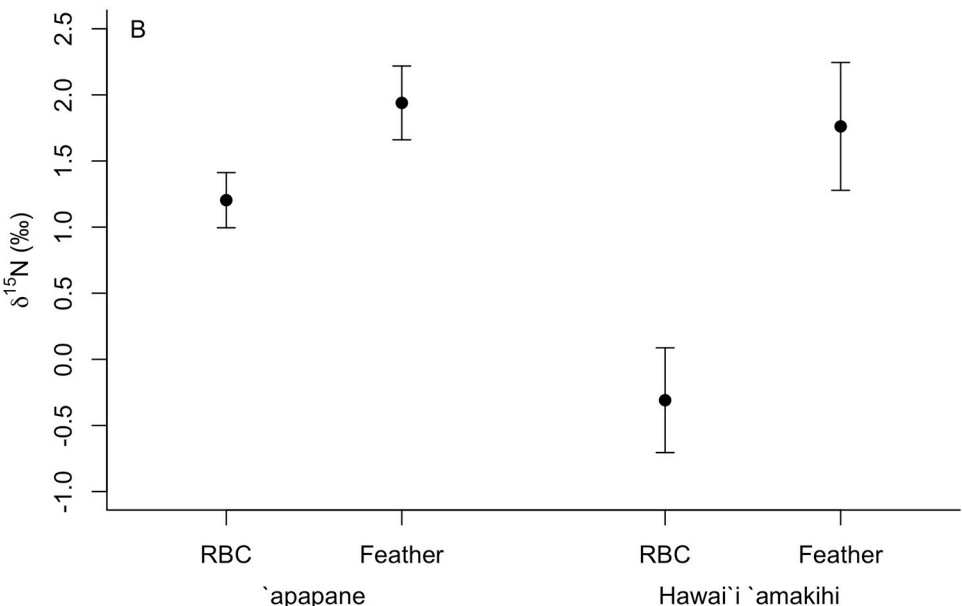

**Fig 4.** (A) Stable carbon isotope ($\delta^{13}$C) and (B) stable nitrogen isotope ($\delta^{15}$N) values (± SE) for red blood cells (RBC) and feathers collected for ʻapapane and Hawaiʻi ʻamakihi in the upper Kaʻū Forest.

extended time period. By examining stable hydrogen isotope values from a feather, RBCs, and blood plasma collected from the same individual during the fall, we documented elevational movements for ʻapapane and Hawaiʻi ʻamakihi between the summer, the period of peak feather molt following breeding, and the time around capture in the fall, represented by the blood plasma sample. RBCs provide information on movement between the two periods given the longer residency time of isotopes in RBCs (6 to 8 weeks) compared to plasma (~3 to 5 days) [32, 33, 37].

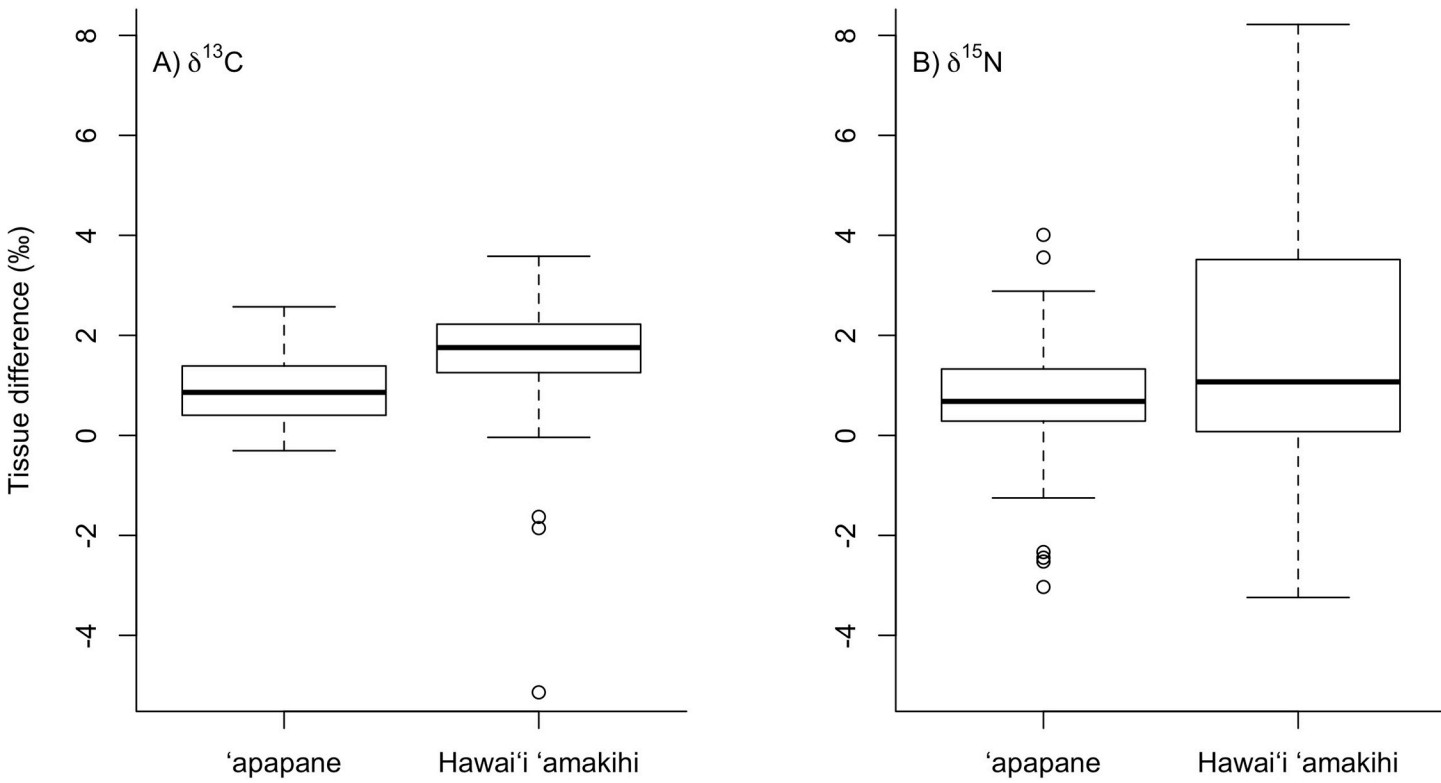

**Fig 5.** Boxplots showing the difference in **(A)** stable carbon isotope ($\delta^{13}C$) and **(B)** stable nitrogen isotope ($\delta^{15}N$) values between feathers and red blood cells collected from ʻapapane and Hawaiʻi ʻamakihi in the upper Kaʻū Forest. Box plot whiskers depict the 10th and 90th percentiles and boxes show the 25th and 75th percentiles with the median value indicated; circles represent outliers.

Consistent with our predictions based on the elevational-migration hypothesis we found that the majority of ʻapapane and Hawaiʻi ʻamakihi captured at high elevation sites in the fall where not year-round residents at the capture locations, but had molted their feathers at lower elevations presumably in the summer after breeding. $\delta^2H$ values of feathers adjusted for tissue-specific discrimination for all ʻapapane and Hawaiʻi ʻamakihi sampled were higher than blood plasma isotope values, and the direction and magnitude of difference between $\delta^2H$ values of feathers and blood plasma indicates that feathers were grown at lower elevations than their capture location. Likewise, estimations of the elevation of feather growth based on the relationship between $\delta^2H_P$ and elevation on the east side of Hawaiʻi Island indicated that the average elevation of feather growth was around 1500m, which is below all three capture locations. Seventy-five percent of the birds captured had an estimated elevation of feather growth below their capture location, with primarily only birds captured at the lowest elevation capture site deviating from this pattern. The molting of feathers at lower elevations than the high elevation capture sites is also consistent with survey data during the breeding season showing the highest densities of ʻapapane and Hawaiʻi ʻamakihi in the Kaʻū Forest around 1500m [43]. Moreover, the closer proximity of $\delta^2H$ values of RBCs to adjusted $\delta^2H$ of feathers (average difference = 10.1‰) compared to plasma (average difference 32.3‰), indicates that birds may have 1) recently migrated to higher elevations, and still retain isotopes from lower elevations among their RBCs, 2) that birds are making daily long-distance movements to forage within both high and low elevation areas, and the isotope signature of RBCs represents an integration of both locations, or 3) some combination of the two scenarios. Visual observations of large numbers

of birds making morning flights above the canopy from lower elevations forests, that were not in bloom, to flowering ʻōhiʻa trees at the high elevation capture sites in the fall (E. Paxton *personal observations*) is consistent with the second scenario. Collectively, the isotope results from multiple tissues provides empirical evidence for seasonal elevational migrations of both ʻapapane and Hawaiʻi ʻamakihi after the breeding season to high elevation sites within the Kaʻū Forest that have heavy ʻōhiʻa bloom.

Surprisingly, we did not find a difference in the propensity of elevational movement between ʻapapane and Hawaiʻi ʻamakihi. Given differences in the foraging strategies of ʻapapane, a nectivore, and Hawaiʻi ʻamakihi, a generalist, we predicted that ʻapapane would be more likely to make elevational movements in search of seasonally variable nectar resources while Hawaiʻi ʻamakihi would be more likely to switch diets when ʻōhiʻa bloom is scarce. However, our results, along with other studies [9, 61, 62] indicate that there may be more overlap in the foraging strategies of these two species, particularly at times when resources are highly concentrated. ʻŌhiʻa accounts for 90% of the trees and shrubs producing nectar in Hawaiian wet forests from sea level to tree line [12, 17], but the bloom of ʻōhiʻa is not uniform in space and time and the timing of peak flowering varies depending on a siteʻs elevation, substrate age, and genetic variation of ʻōhiʻa varieties present [14, 15, 17]. High elevation ʻōhiʻa varieties *polymorpha* and *incana* bloom in fall and winter, whereas lower elevation varieties such as *glaberrima* have peak bloom primarily in spring [15, 17, 46]. Differences in the timing of bloom by variety and elevation creates spatially and temporaly variable distributions of bloom that may drive elevational migration. Indeed, the mature stature ʻōhiʻa forest below our capture sites had virtually no bloom and the forest was quiet with little bird activity detected. In contrast, the low stature ʻōhiʻa patches where we captured birds in the fall had heavy bloom and high densities of birds, which were evident by both sight and auditory detection (E. Paxton *personal observations*). The high energy content of nectar compared to arthropods [63], may make long distant flights to track nectar resources across the landscape beneficial from an energetic consideration (e.g., [11]) for not only true nectivores like ʻapapane and ʻiʻiwi (*Drepanis coccinea*), but also Hawaiʻi ʻamakihi, especially when high volumes of nectar are concentrated in a particular area like the high elevation capture sites in this study. ʻApapane are conspicous when moving long distances, flying above the canopy, whereas Hawaiʻi ʻamakihi are rarely seen flying above the canopy and likely move within or below the forest canopies (E. Paxton *personal observations*). Differences in the conspicuousness of the two speciesʻ flight patterns potentially associated with different foraging strategies may account for the perception that Hawaiʻi ʻamakihi do not move as much as ʻapapane. A better understanding of the consistency of bloom patterns across time within the Kaʻū Forest and other forests in Hawaiʻi would help to shed light on the mechanisms underlying the patterns found in this study.

The incorporation of carbon and nitrogen stable isotopes provides further evidence for the use of nectar resources by Hawaiʻi ʻamakihi at the high elevation capture sites in the fall. Hawaiʻi ʻamakihi and ʻapapane both had significant differences between $\delta^{13}C$ and $\delta^{15}N$ values of feathers and RBCs. However, the average difference between tissue types for ʻapapane was less than 2‰, indicating that for the majority of ʻapapane the change in isotope values between feathers and RBCs most likely represents only differences in tissue discrimination of isotopes [36, 55, 56], and not a shift in diet between seasons. In contrast, the majority of Hawaiʻi ʻamakihi had differences in isotope values between feathers and RBCs greater than 2‰, suggesting differences in isotope values between feathers and RBCs most likely represents a shift in diet between seasons. The high variability in $\delta^{13}C$ and $\delta^{15}N$ values of Hawaiʻi ʻamakihi also indicated that they had a more diverse diet across trophic levels than ʻapapane, particularly during the post-breeding period of feather molt. However, the greater depletion of N in RBC samples compared to feather samples of Hawaiʻi ʻamakihi indicated a greater reliance in the fall on

nectar which is more depleted in N [34, 64]. The incorporation of $\delta^{13}C$ and $\delta^{15}N$ from plasma or breath samples in future studies would help to elicudate the role of nectar in the diet at the time of capture. In addition, tissue-specific discrimination factors for our study species or a comparable nectarivore species would allow for a more precise understanding of changes in diet between seasons.

The Hawaiian Archipelago, and particularly Hawai'i Island, is ideal for studying elevational movements with $\delta^2H$ because of large elevational gradients (e.g., 0–4000 m) that occur across small geographic areas, resulting in a strong gradient of $\delta^2H_p$ values that are driven by changes in elevation and not latitude. The rate of change in $\delta^2H_p$ across elevations in Hawai'i (~1.8‰ per 100m) is consistent with other mountainous systems (e.g. Appalachian and Ecuadorian Andes Mountains) [24, 28, 31] and global patterns of precipitation (e.g., IAEA value for Hilo; [57]). However, unlike other tropical systems that have large seasonality in rainfall patterns (e.g., wet and dry seasons), which can result in different isotopic values between seasons [27], rainfall patterns on the Hawaiian Islands are largely driven by trade-winds resulting in consistent annual and seasonal $\delta^2H$ values across elevations [44, 65]. While storm systems during the winter months, when trade-winds are less frequent, can sometimes result in lower $\delta^2H_p$ values than expected, there was not an increase in storm events (>50mm rainfall in one event; definition given by [57]) during the time period of the study (NOAA National Climate Data Center for Hilo, Hawai'i, Network ID: GHCND:USW00021504).

Mobile animals such as birds can move across the landscape irespective of jurisdictional boundaries, which creates unique managment and conservation problems. Much of Ka'ū Forest is managed by Hawai'i Division of Forestry and Wildlife; however, the upper portion of the forest, where our study sites were located, is within the boundaries of Hawai'i Volcanoes National Park. Our study indicates that birds in the Ka'ū Forest move between the two reserves, and may be dependent on two different management entities, highlighting the importance of understaning movement of birds across the landscape, and how that movement connects different spatial areas over time. Ultimately, conservation of Hawaiian forest birds such as the 'apapane and Hawai'i 'amakihi may depend on the joint-management of lands under different owerships to ensure that habitat quality and protection is sufficient for the birds across the annual cycle.

## Acknowledgments

Darcy Hu (National Park Service) was a key supporter of the project, and helped shape study design. Field work was conducted by Nolan Lancaster, Sonia Levitz, and Keith Burnett. We thank Hawai'i Volcanoes National Park for land access.

## Author Contributions

**Conceptualization:** Kristina L. Paxton, Jeffery F. Kelly, Eben H. Paxton.

**Data curation:** Sandra M. Pletchet.

**Formal analysis:** Kristina L. Paxton.

**Funding acquisition:** Eben H. Paxton.

**Writing – original draft:** Kristina L. Paxton.

**Writing – review & editing:** Kristina L. Paxton, Jeffery F. Kelly, Eben H. Paxton.

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
