## [Decision Letter · Decision Letter 0]

23 Mar 2020

PONE-D-19-34295

Stable isotope analysis of multiple tissues from Hawaiian honeycreepers indicates elevational movement

PLOS ONE

Dear Dr. Paxton,

Thank you for submitting your manuscript to PLOS ONE. After careful consideration, we feel that it has merit but does not fully meet PLOS ONE’s publication criteria as it currently stands. Therefore, we invite you to submit a revised version of the manuscript that addresses the points raised during the review process.

The main points that both reviewers make is that you need to find discrimination factors between blood and feathers for these species. They also include many other useful comments. Please be sure to look at Reviewer 1 comments in the attached word file. 

We would appreciate receiving your revised manuscript by May 07 2020 11:59PM. To enhance the reproducibility of your results, we recommend that if applicable you deposit your laboratory protocols in protocols.io, where a protocol can be assigned its own identifier (DOI) such that it can be cited independently in the future. For instructions see: http://journals.plos.org/plosone/s/submission-guidelines#loc-laboratory-protocols

We look forward to receiving your revised manuscript.

Kind regards,

David P. Gillikin, Ph.D.

Academic Editor

PLOS ONE

Journal Requirements:

Reviewers' comments:

Reviewer's Responses to Questions

**Comments to the Author**

1. Is the manuscript technically sound, and do the data support the conclusions?

Reviewer #1: Yes

Reviewer #2: Partly

2. Has the statistical analysis been performed appropriately and rigorously? 

Reviewer #1: Yes

Reviewer #2: No

3. Have the authors made all data underlying the findings in their manuscript fully available?

Reviewer #1: Yes

Reviewer #2: Yes

4. Is the manuscript presented in an intelligible fashion and written in standard English?

Reviewer #1: Yes

Reviewer #2: Yes

5. Review Comments to the Author

Reviewer #1: I enjoyed reading this manuscript very much. The grammar and clarity made it easy to review. The authors have used tissue stable isotopes in a fascinating, appropriate, and somewhat novel way. My primary concern is with the application of a conversion equation for hummingbirds that may not be appropriate here and, specifically, may have artificially inflated the support for the elevation-migrant hypothesis. When this problem is dealt with the manuscript is otherwise a worthy contribution to its field.

Reviewer #2: This paper uses stable isotopes of mainly hydrogen to explore the use of different elevations of two species of Hawaiian birds within a montane environment. The paper is well written and well thought out. Although the use of stable isotopes of hydrogen has been used in this context previously, and the authors present interesting results, they do not account for the discrimination factors between tissues and therefore the results are not interpreted correctly. The authors need to find discrimination factors between blood and feathers for these species they are interested in, or in a species that is somewhat similar to approximate it. After accounting for the discrimination factors between tissues (for each stable isotope separately, it will differ for carbon, hydrogen and nitrogen) then the authors can proceed with the analyses as they had. This may change their results of elevation pattern use for the two species and also differences in dietary niche. The authors mention discrimination factors in the discussion (lines 288-292), so they are clearly aware of this, however I think it is crucial that these are incorporated in their analyses and adding this as a discussion point is insufficient.

Below I make other recommendations with which to improve this manuscript.

Line 64: define the two terms frugivorous and nectivorous. Within the discussion also include some approximation of how these two diets may differ isotopically.

Line 72: which other feeding guilds are you referring to? Please provide an example or two.

Line 78: Do you mean that the birds “tracked the timing of the flowering ‘ōhi’a” ? If so, please change wording accordingly. I think this introduction would benefit from a sentence detailing when this tree flowers at which elevation and if there are other trees that flower in this area and when. It is a bit unclear to those that might not be familiar with this landscape.

Line 102: Define the term “orographically” or better yet write this sentence without such jargon. This paper will primarily be read by bird researchers who are unlikely to know such climatic terms.

Line 103-105: Stable isotopes of nitrogen is correlated with trophic level (becoming more and more enriched with the increase of trophic level) not hydrogen. Hydrogen is fairly insensitive to trophic level. There are a lot of papers out there outlining this, but these two are also.

G. Bowen et al 2005 Global application of stable hydrogen and oxygen isotopes to wildlife forensics. Oecologia 143: 337–348

K. Hobson et al 1994 Using stable isotopes to determine seabird trophic relationships. J. Animal Ecol. 63(4):786-798

Line 131-134: Here the authors even explicitly state that discrimination factors are important to consider, but then they’re not included in the analyses. Would suggest rewording this sentence to end with “as long as comparisons are made while also incorporating tissue-specific discrimination factors”. It is necessary to account for discrimination factors in your analyses. In line 146 you again write about discrimination values, yet don’t actually account for them as far as I can tell. Without accounting for discrimination between tissues you cannot be certain that any differences you see are due to elevation.

Line 135: why the windward side? Please explain. This could also be a good space to include details about the flowering cycle of the primary source of nectar.

Line 137: A sentence detailing the annual-cycle and timing of each stage is necessary here. When do the birds molt? Migrate to where? Breed?

Line 172: How come birds weren’t also sampled at lower elevations? Or are there no birds present at lower elevations at all?

Line 196: Could be helpful to include a figure outlining tissues and what time period they would reflect.

Line 201: Were plasma and RBC not lipid extracted? Why not? For stable isotopes of hydrogen it doesn’t really matter, but plasma is highly enriched in lipids, and therefore not lipid extracting can influence your stable isotopes of carbon values.

Line 208: Are you sure this is caribou and not cow? Should include references for these standards, and also the variation throughout your analyses of the standards (which is the whole reason they are included anyway, to ensure the stable isotope analysis is accurate throughout the analyses and doesn’t trend in one direction or the other indicating that the Mass Spec isn’t working). You should include average value +/- standard deviation for each standard and how often you analysed the standards throughout the analyses (often its every 5th or 10th sample). There should have also been duplicates of at least each 10th sample you analysed to ensure that the samples were homogenous.

Line 242: How come there are so many fewer samples for C/N? (guessing that the numbers for C are the same as for N, but you should also include that explicitly in your sample size here).

All of the analyses in the results sections need to be redone after accounting for discrimination factors between tissues. I’m sorry, this is always difficult to hear, but I think your paper will be much better after accounting for this. Otherwise you cannot be confident that the differences you see are actually because of elevational differences between different stages during the annual cycle. Other than this oversight, the statistical analyses are quite well done, and easy to follow.

Lin 289-292: Its obvious you are aware of these differences. You can pretty much remove this section once the analyses are re-done. Over half of the differences you saw in elevation are actually due to discrimination factors. Which means that the calculations you made for use of different elevations is actually incorrect, likely half of what you calculated.

Line 298-305: Not that I’m recommending this for this paper, but one alternative method you could consider for future exploration of use of different elevations is using VHF tags and a motus type network (widely used in eastern Canada/US). They are generally quite light weight and are a good alternative to GPS tags. Just a thought!

Line 332: Could these flight patterns also be due to differences in foraging strategies?

Line 343-345: You didn’t look at this in plasma? How come? (possible I missed this explanation in the methods).

Line 368: typo. “different” not “differnt”

6. PLOS authors have the option to publish the peer review history of their article (what does this mean?). If published, this will include your full peer review and any attached files.

Reviewer #1: Yes: Cameron Nordell

Reviewer #2: No

---

## [Author Response · Author response to Decision Letter 0]

19 May 2020

Dear Dr. Gillikin,

Please accept the revised manuscript (PONE-D-19-34295) titled: “Stable isotope analysis of multiple tissues from Hawaiian honeycreepers indicates elevational movement” for publication in PLOS ONE. We thank the two reviewers for their thorough and insightful comments. We have responded to the requested revisions, which have improved the manuscript. Please see below for more detailed responses (in blue) to comments made by the reviewers (in black). The page and line numbers indicated in our responses refer to line numbers in the revised manuscript with track changes. We hope you will find that we have addressed the comments of the reviewers and thank you for your consideration of the revised manuscript for publication in PLOS ONE.

Sincerely,

Kristina Paxton, PhD

University of Hawai‘i at Hilo

Response to Reviewers 

Reviewer #1: Cameron Nordell

I enjoyed reading this manuscript very much. The grammar and clarity made it easy to review. The authors have used tissue stable isotopes in a fascinating, appropriate, and somewhat novel way. My primary concern is with the application of a conversion equation for hummingbirds that may not be appropriate here and, specifically, may have artificially inflated the support for the elevation-migrant hypothesis. When this problem is dealt with the manuscript is otherwise a worthy contribution to its field.

Grammatical Comments:

Line 38: In the abstract I believe you use the word “where” instead of “were” 

Lines 201-202: You used the word “powered” instead of “powdered” twice.

Response: All grammatical errors were corrected

Specific Comments

Line 95-96

I find it imprecise to say that isotopes are incorporated into animal tissues through trophic level interactions. You might more closely reflect the relationship of animal tissues to isotopes by stating that they are incorporated through biochemical processes

(e.g. Breathing is, in part, responsible for the isotope ratios in an organism’s tissues).

Line 103-104

As with my comment on Line 95, I suggest that the δ2H isotope ratios in animal tissues reflect their environment not singularly because of trophic interactions. 

You might also reconsider the use of the word “highly” to describe the relationship between tissues and δ2H isotope ratios, given the many confounding factors that obscure this relationship (e.g. evapotranspiration of environmental or biological water differs among individuals and perhaps small-scale location). I think the theme of quantifying the factors that obscure δ2HP and δ2H in animal tissues, in particular, is common in recent papers.

You might also consider the addition “expected to reflect the isotopic signatures of the elevation of feeding when the tissue was grown”

Response: As suggested we updated the text in the introduction to better represent how stable isotopes are incorporated into animal tissues and replaced ‘trophic interactions’ with ‘biochemical processes’. [lines 104, 116]

Line 157

I’d suggest that “greater inter-tissue variation” might better capture your meaning here.

Response: As suggested text was changed from ‘greater inter-tissue differences’ to ‘greater inter-tissue variation’ [line 178]

Line 189

Were tail feather selected for any reason? The outer tail feathers are molted last in many species. Is there a possibility to mis-classify some lower-elevation migrants as higher-elevation locals by sampling feathers that were recently replaced (i.e. outer tail feathers, the last replaced) at higher elevations? If this went unaccounted for, you might actually find stronger support for your elevational-migrant hypothesis.

Further, were the tail feathers counted from the center to determine the feather taken was the true outer feather and not the outermost that is not actively being replaced? Failure to account for this could introduce additional unwanted noise into your data.

Response: We choose tail feathers because they are a standard feather used for stable isotope analysis. For each bird captured, we assessed for the presence of molt and did not include birds that were actively molting tail feathers. Because all birds were sampled the same, any bias would be the same for all individuals. [line 215]

Lines 266-267. 

The cutoff of 2‰ used to assert that differences between ‘apapane tissues owe to differential discrimination while in differences between tissues amakihi’ are due to diet changes seems arbitrary. Without stronger apriori assertions about 2‰ it seems equally valid to conclude that amakihi’ demonstrated a greater seasonal shift in diet than ‘apapane. Or that the isotope discrimination between the two tissues was greater in ‘apapane than amakihi’. Further, I’d suggest this is best left for the discussion rather than the results section.

Response: As suggested we moved the discussion of the results to the discussion section. [443-451] In addition, in the methods section we provided text to support our use of 2‰ as a discrimination factor between RBCs and feathers to assess large-scale changes in diet. [lines 280-286]

Line 272-274

Does the larger variation in δ13C not also suggest a more diverse diet? Regardless, this sentence might be best positioned in your discussion section.

Response: We agree with the reviewer and removed the text from the results section. In the discussion section we included text discussing support for ’amakihi having a more diverse diet than ‘apapane based on both δ13C and δ15N. [line 450-453]

Lines 352-355

I think this is an important distinction for your study area. Good inclusion.

Response: Thank you

General Comments

A) I assume all birds used in your study were AHY individuals? Is this simple to determine using plumage differences between AHY and HY? Other studies have found significant isotope ratio differences between AHY and HY individuals at the same breeding site. (Hache et al. 2012 - Assigning birds to geographic origin using feather hydrogen isotope ratios (δ2H): importance of year, age, and habitat)

Response: Yes there are distinct plumage differences between HY and AHY birds. All birds were aged and sexed using plumage, breeding characteristics, and size as described in Paxton et al. 2016 – Aging and sexing guide to the forest birds of Hawai’i Island. Over 90% of the birds in our study were AHY. We did not have a large enough sample size to test for differences between AHY and HY birds. [lines 212-215]

B) The most important concern with the manuscript: I am suspicious of the application of a conversion equation that describes the relationship between δ2HP and δ2Hf in hummingbirds to your study species. Elsewhere, among closely related species breeding in the same area, species-specific equations to convert δ2HP to δ2Hf appear to be necessary (Nordell et al. 2016 - Within-Site Variation in Feather Stable Hydrogen Isotope (δ2 Hf) Values of Boreal Songbirds: Implications for Assignment to Molt Origin).

I am no biochemist, but the particularly extreme metabolism of hummingbirds might lead to biochemical isotope discrimination at significantly greater rates than perhaps any other species.

I am not sure whether the fast hummingbird metabolism would incorporate light hydrogen or deuterium (heavy hydrogen) preferentially in metabolic processes. The worst case scenario for your manuscript would be that deuterium is more concentrated in hummingbird tissues than in your study species, which would lead you to predict δ2H values in your species originated at lower elevations than they truly did. In that case, you might find considerably less support for your elevation-migrant hypothesis.

I suggest you explore the literature to determine whether the use of the conversion equation for hummingbirds is valid (I don’t think the fact that they are nectivores is justification enough) and, if necessary, utilize a conversion equation more representative of your species (e.g. other fringilidae).

Response: We agree with the reviewer that it would be best to use a species-specific conversion equation. However, we did not have a species-specific conversion equation for our study species or any closely related species, so we used the best approximation available. As nectarivores, hummingbirds and Hawaiian honeycreepers receive most of their hydrogen from plant-derived water and plant-derived carbohydrates through consumption of large quantities of nectar. Thus, hummingbirds likely provide the best approximation of the relationship between feather and precipitation �2H values for Hawaiian honeycreepers. In the text, we discuss the need for species-specific equations, particularly for Hawaiian nectarvores. [lines 297-306]

Reviewer #2:

This paper uses stable isotopes of mainly hydrogen to explore the use of different elevations of two species of Hawaiian birds within a montane environment. The paper is well written and well thought out. Although the use of stable isotopes of hydrogen has been used in this context previously, and the authors present interesting results, they do not account for the discrimination factors between tissues and therefore the results are not interpreted correctly. The authors need to find discrimination factors between blood and feathers for these species they are interested in, or in a species that is somewhat similar to approximate it. After accounting for the discrimination factors between tissues (for each stable isotope separately, it will differ for carbon, hydrogen and nitrogen) then the authors can proceed with the analyses as they had. This may change their results of elevation pattern use for the two species and also differences in dietary niche. The authors mention discrimination factors in the discussion (lines 288-292), so they are clearly aware of this, however I think it is crucial that these are incorporated in their analyses and adding this as a discussion point is insufficient.

Below I make other recommendations with which to improve this manuscript.

Line 64: define the two terms frugivorous and nectivorous. Within the discussion also include some approximation of how these two diets may differ isotopically.

Response: As suggested we defined frugivorous and nectivorous in the text. [line 68]

Line 72: which other feeding guilds are you referring to? Please provide an example or two.

Response: As suggested we provided examples of the alternative feeding guilds that we referred to in the introduction. [line 76]

Line 78: Do you mean that the birds “tracked the timing of the flowering ‘ōhi’a” ? If so, please change wording accordingly. I think this introduction would benefit from a sentence detailing when this tree flowers at which elevation and if there are other trees that flower in this area and when. It is a bit unclear to those that might not be familiar with this landscape.

Response: As suggested we have added text in the introduction that provides more details about the phenology of ‘ōhi‘a and its importance as a nectar resource to Hawaiian honeycreepers. [lines 81 – 87]

Line 102: Define the term “orographically” or better yet write this sentence without such jargon. This paper will primarily be read by bird researchers who are unlikely to know such climatic terms.

Response: As suggested we have removed the word “orographically” and replaced the text with “over a mountain range” to make the text more understandable to a broad audience. [line 114]

Line 103-105: Stable isotopes of nitrogen is correlated with trophic level (becoming more and more enriched with the increase of trophic level) not hydrogen. Hydrogen is fairly insensitive to trophic level. There are a lot of papers out there outlining this, but these two are also.

G. Bowen et al 2005 Global application of stable hydrogen and oxygen isotopes to wildlife forensics. Oecologia 143: 337–348

K. Hobson et al 1994 Using stable isotopes to determine seabird trophic relationships. J. Animal Ecol. 63(4):786-798

Response: To clarify the text we have removed “trophic level interactions” and replaced it with “biochemical processes” to accurately reflect the relationship with between stable hydrogen isotopes in animal tissues and precipitation. [lines 115-117]

Line 131-134: Here the authors even explicitly state that discrimination factors are important to consider, but then they’re not included in the analyses. Would suggest rewording this sentence to end with “as long as comparisons are made while also incorporating tissue-specific discrimination factors”. It is necessary to account for discrimination factors in your analyses. In line 146 you again write about discrimination values, yet don’t actually account for them as far as I can tell. Without accounting for discrimination between tissues you cannot be certain that any differences you see are due to elevation.

Response: We have rewritten the text in the introduction as suggested.[lines 146-149]

Line 135: why the windward side? Please explain. This could also be a good space to include details about the flowering cycle of the primary source of nectar.

Response: In the text we replaced ‘windward’ with ‘east’ to clarify that this text is just referring to the location of the study area on Hawai‘i island. [lines 150-152] We added more text to the introduction, in an earlier paragraph, about the phenology of ‘ōhi‘a flowering to help readers not familiar with the study system. [lines 81-87]

Line 137: A sentence detailing the annual-cycle and timing of each stage is necessary here. When do the birds molt? Migrate to where? Breed?

Response: We provided more detailed information about the annual-cycle and timing of breeding and molt in the text of the introduction [line 153-155] and methods [lines 226-230]. 

Line 172: How come birds weren’t also sampled at lower elevations? Or are there no birds present at lower elevations at all?

Response: We added text to the methods section to indicate that birds were sampled at lower elevations, but no birds were captured most likely because densities of birds were so low. [lines 203-205]

Line 196: Could be helpful to include a figure outlining tissues and what time period they would reflect.

Response: We did not include a figure, but clearly outlined the time period that each tissue represents in the text. [lines 225-231]

Line 201: Were plasma and RBC not lipid extracted? Why not? For stable isotopes of hydrogen it doesn’t really matter, but plasma is highly enriched in lipids, and therefore not lipid extracting can influence your stable isotopes of carbon values.

Response: We did not extract lipids from plasma and RBCs due to the low proportion of lipids in bird blood. However, plasma samples were only included for stable hydrogen analysis and not carbon and nitrogen analysis. [lines 236-237]

Line 208: Are you sure this is caribou and not cow? Should include references for these standards, and also the variation throughout your analyses of the standards (which is the whole reason they are included anyway, to ensure the stable isotope analysis is accurate throughout the analyses and doesn’t trend in one direction or the other indicating that the Mass Spec isn’t working). You should include average value +/- standard deviation for each standard and how often you analysed the standards throughout the analyses (often its every 5th or 10th sample). There should have also been duplicates of at least each 10th sample you analysed to ensure that the samples were homogenous.

Response: We have added to the Methods section all information relating to the standards and their precision. [lines 255-258, lines 261-266]

Line 242: How come there are so many fewer samples for C/N? (guessing that the numbers for C are the same as for N, but you should also include that explicitly in your sample size here).

Response: Samples sizes differed between hydrogen and carbon/nitrogen analysis because of the differences in the amount of blood collected from each bird. We prioritized the stable hydrogen isotope analysis when blood volumes were too low for both analyses. We have included text in the methods section to clarify this point. [lines 215-222]

All of the analyses in the results sections need to be redone after accounting for discrimination factors between tissues. I’m sorry, this is always difficult to hear, but I think your paper will be much better after accounting for this. Otherwise you cannot be confident that the differences you see are actually because of elevational differences between different stages during the annual cycle. Other than this oversight, the statistical analyses are quite well done, and easy to follow.

Lin 289-292: Its obvious you are aware of these differences. You can pretty much remove this section once the analyses are re-done. Over half of the differences you saw in elevation are actually due to discrimination factors. Which means that the calculations you made for use of different elevations is actually incorrect, likely half of what you calculated.

Response: Given the lack of tissue-specific discrimination factors for our study species we originally choose to not adjust feather values prior to analysis, but to instead look at the magnitude of differences between tissues and assess whether the difference observed exceeded the offset expected based on general tissue discrimination values for hydrogen. However, in light of the reviewers comment we have re-run the general linear mixed model for stable hydrogen isotopes with feather values adjusted to account for tissue-specific discrimination between feathers and blood using the average tissue discrimination factor between feathers and blood calculated from an experiment with house sparrows, the species most closely related to Hawaiian honeycreepers with tissue-specific discrimination values. 

[lines 273-280] 

The new analysis does not change the overall conclusions of our study and still supports the elevational-migration hypothesis that the majority of birds captured at the high elevation sites where not year-round residents, but had molted their feathers at lower elevations. The average difference in adjusted �2H values of feathers to plasma is 32.3‰, while the average difference in adjusted �2H of feathers to RBCs is 10.1‰. [lines 317-321]

However, we did not adjust feather values for the carbon and nitrogen analysis because tissue discrimination factors for carbon and nitrogen are highly sensitive to diet (e.g. the amount of protein in the diet) and discrimination factors have not been established for a nectarivore. In addition, published discrimination factors have primarily been calculated for feathers and whole blood, but not RBCs. Instead, similar to the approach of Podlesak et al. 2005 we examined only general changes in diet by considered differences in �13C and �15N values between feathers and RBCs greater than 2‰ to be indicative of a change in diet. This approach allows us to make a general interpretation of whether diets may have changed between time periods. [lines 280-286] We have also acknowledged in the discussion that tissue discrimination factors are needed for our study species or nectivores in general to make this analysis more robust. [lines 456-458]

Line 298-305: Not that I’m recommending this for this paper, but one alternative method you could consider for future exploration of use of different elevations is using VHF tags and a motus type network (widely used in eastern Canada/US). They are generally quite light weight and are a good alternative to GPS tags. Just a thought!

Response: Good suggestion for future studies

Line 332: Could these flight patterns also be due to differences in foraging strategies?

Response: Yes, differences in flight patterns could be a result of differences in foraging strategies. [lines 433-435]

Line 343-345: You didn’t look at this in plasma? How come? (possible I missed this explanation in the methods).

Response: In the methods we indicate that only RBCs, not plasma was run for carbon and nitrogen analyses. We have added text in the methods to clarify that the volume of blood collected did not permit the analysis of plasma for both hydrogen and carbon/nitrogen analyses. [lines 215-222]

Line 368: typo. “different” not “differnt”

Response: We have fixed the typo [line 485]

---

## [Decision Letter · Decision Letter 1]

23 Jun 2020

Stable isotope analysis of multiple tissues from Hawaiian honeycreepers indicates elevational movement

PONE-D-19-34295R1

Dear Dr. Paxton,

We’re pleased to inform you that your manuscript has been judged scientifically suitable for publication and will be formally accepted for publication once it meets all outstanding technical requirements. Congratulations!

Kind regards,

David P. Gillikin, Ph.D.

Academic Editor

PLOS ONE

Additional Editor Comments (optional):

Reviewers' comments:

Reviewer's Responses to Questions

**Comments to the Author**

1. If the authors have adequately addressed your comments raised in a previous round of review and you feel that this manuscript is now acceptable for publication, you may indicate that here to bypass the “Comments to the Author” section, enter your conflict of interest statement in the “Confidential to Editor” section, and submit your "Accept" recommendation.

Reviewer #1: All comments have been addressed

Reviewer #2: All comments have been addressed

2. Is the manuscript technically sound, and do the data support the conclusions?

Reviewer #1: Yes

Reviewer #2: Yes

3. Has the statistical analysis been performed appropriately and rigorously? 

Reviewer #1: Yes

Reviewer #2: Yes

4. Have the authors made all data underlying the findings in their manuscript fully available?

Reviewer #1: Yes

Reviewer #2: Yes

5. Is the manuscript presented in an intelligible fashion and written in standard English?

Reviewer #1: (No Response)

Reviewer #2: Yes

6. Review Comments to the Author

Reviewer #1: The authors have fully addressed my comments and the manuscript is a worthy contribution to the field.

Reviewer #2: The authors have adequately addressed all comments included in the previous review process. The manuscript is well written, easy to follow and statistical analyses have been conducted in a sound manner. Re-running analyses after receiving a review is never an easy task, and the authors have made adequate efforts to address these comments and amended their manuscript appropriately.

7. PLOS authors have the option to publish the peer review history of their article (what does this mean?). If published, this will include your full peer review and any attached files.

Reviewer #1: Yes: Cameron J. Nordell

Reviewer #2: Yes: Rolanda J Steenweg

---

## [Editor Report · Acceptance letter]

30 Jun 2020

PONE-D-19-34295R1 

Stable isotope analysis of multiple tissues from Hawaiian honeycreepers indicates elevational movement 

Dear Dr. Paxton:

I'm pleased to inform you that your manuscript has been deemed suitable for publication in PLOS ONE. Congratulations! Your manuscript is now with our production department. 

Kind regards, 

on behalf of

Dr David P. Gillikin 

Academic Editor

PLOS ONE